# Chronic Wasting Due to Liver and Rumen Flukes in Sheep

**DOI:** 10.3390/ani11020549

**Published:** 2021-02-19

**Authors:** Alexandra Kahl, Georg von Samson-Himmelstjerna, Jürgen Krücken, Martin Ganter

**Affiliations:** 1Institute for Parasitology and Tropical Veterinary Medicine, Freie Universität Berlin, Robert-von-Ostertag-Str. 7-13, 14163 Berlin, Germany; samson.georg@fu-berlin.de (G.v.S.-H.); juergen.kruecken@fu-berlin.de (J.K.); 2Clinic for Swine and Small Ruminants, Forensic Medicine and Ambulatory Service, University of Veterinary Medicine Hannover, Foundation, Bischofsholer Damm 15, 30173 Hannover, Germany; martin.ganter@tiho-hannover.de

**Keywords:** sheep, trematodes, flukes, infection, wasting, emaciation

## Abstract

**Simple Summary:**

Chronic wasting in sheep is often related to parasitic infections, especially to infections with several species of trematodes. Trematodes, or “flukes”, are endoparasites, which infect different organs of their hosts (often sheep, goats and cattle, but other grazing animals as well as carnivores and birds are also at risk of infection). The body of an adult fluke has two suckers for adhesion to the host’s internal organ surface and for feeding purposes. Flukes cause harm to the animals by subsisting on host body tissues or fluids such as blood, and by initiating mechanical damage that leads to impaired vital organ functions. The development of these parasites is dependent on the occurrence of intermediate hosts during the life cycle of the fluke species. These intermediate hosts are often invertebrate species such as various snails and ants. This manuscript provides an insight into the distribution, morphology, life cycle, pathology and clinical symptoms caused by infections of liver and rumen flukes in sheep. Furthermore, we address the diagnosis, treatment and prevention measures, as well as the current knowledge about genomic resources.

**Abstract:**

Grazing sheep and goats are constantly exposed to helminth infections in many parts of the world, including several trematode species that causes a range of clinical diseases. The clinical picture of flukes is dependent upon the organs in which they develop and the tissues they damage within the respective organs. Accordingly, infections with the common liver fluke *Fasciola hepatica*, which, as juvenile worm migrates through the liver parenchyma for several weeks, may be associated with hepatic disorders such as impairment of carbohydrate, protein and fat metabolism, followed by chronic wasting. In contrast, the lancet fluke *Dicrocoelium dendriticum*, which does not exhibit tissue migration and thus does not lead to major tissue damage and bleeding, also does not lead to significant clinical symptoms. Rumen flukes such as *Cotylophoron daubneyi* cause catarrhal inflammation during their migration through the intestinal and abomasal epithelium during its juvenile stages. Depending on the infection intensity this may result in a range of clinical symptoms including diarrhoea, inappetence or emaciation. In this review, we aim to provide an update on the current knowledge on flukes particularly concerning the clinical relevance of the most important fluke species in sheep.

## 1. Introduction

Digenea are platyhelminths of the class Trematoda and have at least two hosts where the definitive host is a vertebrate while the first intermediate host is most often a mollusk (frequently a gastropod) or rarely a polychaet. Various invertebrate and vertebrate species may serve as intermediate and paratenic hosts [1]. Accordingly, fluke prevalence is highly dependent on the macro- and microclimatic conditions that enable survival and interactions with specific intermediate hosts [2,3]. Grazing animals, particularly sheep, are highly susceptible to various fluke species. In Europe, these include the common liver fluke *Fasciola hepatica*, the lancet fluke *Dicrocoelium dendriticum* and various rumen fluke species belonging to the genera *Paramphistomum* or *Calicophoron*. Fluke infections in sheep cause a wide variety of clinical symptoms depending on the specific organs associated with their development and manifestation [4]. While chronic wasting is not a key clinical sign for all of the above-mentioned fluke infections, it is considered a condition that always arises depending on the different factors within the parasite–host interaction. This paper aims at providing an overview on the current knowledge of fluke infections for the development of this often-occurring clinical picture.

## 2. Fasciolidae with Emphasis on the Common Liver Fluke

Members of the family *Fasciolidae*, also known as the common liver flukes, are trematodes that parasitize in the liver of particular herbivore mammals, especially in ruminants. Infections of other species as definitive hosts (e.g., equines, pigs, hares, dogs and rats) [5] are also possible but occur rarely. The highest pathogenic impact is in sheep, where fasciolosis may cause high morbidity and mortality leading to economic loss of EUR 635 million per year just in Europe due to production losses and treatment costs [6].

Furthermore, not only animals but also humans are at risk of infection by consuming contaminated freshwater plants carrying the infectious metacercariae of the parasite. Thus, making fasciolosis a zoonotic disease. Human fasciolosis particularly occurs in developing countries [7], but in animals, the disease is common in many parts of the world constituting a serious animal health problem.

### 2.1. Distribution of and Anatomical Differences between Fasciola Species

There are two different species of the parasite that differ in their spatial distribution. *Fasciola hepatica* is distributed globally in temperate and subtropical areas in Europe, America, Asia, Australia and some parts of Africa, whereas *Fasciola gigantica* exists only in tropical areas of Africa and Asia [8].

*Fasciola hepatica* (Figure 1) is a grey-brownish, leaf-shaped fluke with an approximate size of about 1.8–5.0 cm in length and 1.0 cm in width [9] (p.172). This species has a conical anterior end, which is delimited by obvious shoulders [10] (p.78). The major parts of the fluke’s bodies serve for reproduction, which results in an extremely high rate of egg production, so that a single fluke may produce up to 25,000 eggs per day [11]. The oval, golden yellow eggs measure 130–145 × 70–90 μm in size and have an operculum [9] (p. 172) (Figure 2).

*Fasciola gigantica* is larger in size than *F. hepatica*, they measure up to 7.5 cm in length and 1.2 cm in width [9] (p. 172). In contrast to *F. hepatica,* the shoulders are not as conspicuous, and the body is more transparent [10] (p. 78). However, in recent publications hybrids between both of these *Fasciola* species have been described as a local phenomenon which most likely happens by interspecies mating in regions where the distribution areas of the species overlap [12].

### 2.2. Life Cycle of Fasciola spp.

Both species have a diheteroxenic life cycle involving freshwater snails of the family Lymnaeidae as intermediate hosts. For *F. hepatica*, the most common intermediate host in Europe is *Galba truncatula*, whereas hosts of *F. gigantica* include *Radix natalensis* and *Radix auricularia* during its life cycle [13].

Infected definitive hosts excrete *Fasciola* eggs with their faeces and thereby depositing them in the environment. Outside of the mammalian host, the eggs finalize their development to the larval stage miracidium. This process is highly dependent on mild temperatures and humidity. The time needed for embryonation varies from six months at 10 °C to eight days at 30 °C. Yet, eggs may maintain their viability at temperatures down to 0 °C for up to two years [14]. After embryonation, the motile miracidium hatches (stimulated by daylight) and swims actively for up to 24 h to find a mollusc as a suitable intermediate host [14]. It penetrates the intermediate host through its foot after locating the snail by chemotaxis. Once inside the snail’s body cavity, the parasite then multiplies at different life cycle stages. The miracidium becomes a sporocyst, which in turn gives rise to up to 200 rediae. Each redia generates the emergence of about 20 cercariae [14]. After a 4–7 weeks infection, the intramolluscan development is complete. A large amount of cercariae erupt from the snail, encysting on vegetation or the surface of water and then enter the infective stage called metacercaria, which may remain infectious for up to one year [14].

Sheep and other definitive hosts ingest the infectious metacercariae by grazing on contaminated pastures or in some instances by drinking out of natural water sources or if they feed on inadequately prepared silage and hay. Infection then initiates with excystation in the rumen, where concentrations of carbon dioxide and warm temperatures stimulate the disintegration of the cyst wall [15]. In the abomasum the enzyme pepsin weakens the cyst wall along with the enzyme trypsin in the small intestine, after which the parasite may finally emerge from the cyst. The immature fluke is now able to penetrate the gut wall and to move through the peritoneal cavity [16]. After reaching the liver and traversing its capsule, the juvenile flukes migrate through the parenchyma for up to six weeks [14]. They settle in the bile ducts as sexually mature adult flukes, where the reproduction cycle begins again, and new eggs are shed via the bile into the faeces. In untreated sheep, the common liver fluke may survive up to several years [15] (p. 78).

A schematic life cycle is depicted on the website of the Center for Disease Control (USA), available online under the following link: www.cdc.gov/parasites/fasciola/biology.html (accessed on 5 February 2021).

### 2.3. Clinical Features of Fasciolosis

The migration through the liver parenchyma of the juvenile flukes causes considerable tissue damage followed by haemorrhage and fibrosis and immunological reactions [17]. According to a study by Rushton and Murray in Scotland in 1977 [18], the highest burden of immature flukes was found in the ventral lobe, where they induced a substantial system of haemorrhagic and necrotic tracks, which later formed conspicuous post-necrotic scars. Furthermore, penetrating flukes provoke portal and hepatic vein stenosis as well as pericellular fibrosis around the triads with hepatocytes either surrounded by collagen fibres or degenerated [18]. The damage is shown in Figure 3.

Adult flukes, which settle in the bile ducts, may cause mechanical obstruction of the bile ducts [17]. As the adults have microscopic spines on their tegument, the movements of the flukes harm the inside of the bile ducts resulting in thickened bile ducts and cholangitis with hyperplasia of the epithelium [19]. The juvenile and the mature *Fasciola*-stages feed by producing enzymes, particularly proteases, which degrade blood and the liver parenchyma [19].

All aspects of liver damage induced by the parasite play a decisive role in the host’s metabolism and impairs the organ’s vital functions. In 2019, a study from Pakistan investigated the pathological impact of *F. hepatica* infections in sheep including measuring the biochemical parameters in the blood [17]. The paper emphasizes the correlation between the degree of pathological hepatic lesions and high activities of the liver enzymes aspartate aminotransferase (AST), alanine aminotransferase (ALT) and alkaline phosphatase (ALP) in serum, which indicate the destruction of the hepatocellular integrity. In contrast to the increased activities of the liver enzymes, the levels of blood urea nitrogen (BUN) and total protein in serum were reduced in *Fasciola*-infected sheep since the liver damage has a clear impact on metabolic pathways such as the urea cycle and protein synthesis. In addition, in this study, the activity of gamma-glutamyltransferase (GGT) was measured. As GGT is a sensitive indicator for bile epithelium damage in sheep, the elevated level in this study insinuated that the abnormal blood parameters resulted from liver degeneration. Furthermore, the study showed that creatinine levels were decreased, which might be associated with muscle loss due to reduced bioavailability of protein [17].

Kozat and Denizhan [20] compared the serum concentrations of total protein, albumin, glucose, cholesterol, triglyceride, high density lipoproteins (HDL), low density lipoproteins (LDL) and very low density lipoproteins (VLDL) values in *Fasciola*-infected sheep with the values of healthy sheep. These parameters turned out to be particularly lower in infected sheep whereas the serum activities of AST, ALT, GGT and lactate dehydrogenase (LDH) were significantly higher compared to the uninfected sheep. The authors underline, that the hepatic tissue damage caused by the parasites also influences the capacity of storing glycogen as an energy reservoir [20]. Consequently, during periods of energy deficit, body fat reserves have been catabolized to form non-esterified fatty acids (NEFAs) causing an abnormal serum lipid pattern. According to Blood et al., *Fasciola* spp. infections reduce fertility, growth rate, and wool production in sheep and the voluntary feed intake decreases, what impairs the efficient use of energy resources [21].

Overall, the compromised liver function leads to disorders of carbohydrate, protein and fat metabolism thereby impacting the health, welfare, and productivity of infected sheep significantly. An older study from Australia by Hawkins and Morris [22] investigated the influence of *F. hepatica* on the change of bodyweight, patch wool growth, fleece weights and feed digestibility in Corriedale sheep. These sheep were artificially infected with different numbers of metacercariae ranging from 50 to 5000. An uninfected control group was used as a reference. The negative changes of these parameters in the sheep correlated with the mean fluke burden. Sheep infected with 1100, 2300 and 5000 metacercariae lost body weight after the initial infection. Between these three groups, there was no significant difference, but these sheep were significantly different from all of the other groups that received a lower infection dose. Groups that received 50, 110, 230 or 500 metacercariae did not lose weight, but weight gain was significantly reduced compared to the uninfected control group. Since the food intake in the groups with a lower infection dose (up to 230 metacercariae) did not decrease and the feed digestibility was constant, the reason for that is most likely a depressed feed conversion efficiency [22]. Regarding the wool growth, the control group showed a particularly higher growth rate than any of the infected groups. That can be explained by the fact, that *Fasciola* spp. consume methionine, an amino acid essential for protein synthesis and wool growth. In the groups of sheep, which received up to 230 metacercariae, no deaths occurred whereas all sheep infected with 1100 or more metacercariae died before the end of the trial after suffering from a chronic course of fascioliasis [22]. Based on this study it can be stated that the impact of a *Fasciola* spp. infection on the health of the animals is strongly dependent on the infection intensity, i.e., the number of flukes concurrently parasitizing in the liver.

An acute or subacute fasciolosis results from the intake of a high quantity of metacercariae in a short time period followed by large numbers of young flukes migrating through the liver parenchyma at the same time. Clinically, sudden deaths of previously healthy animals may occur in serious cases as a consequence of hepatic haemorrhages. Other symptoms are reduced food uptake, lethargy, pale or icteric mucous membranes, abdominal pain and dyspnoea [23]. In acute cases the migrating young flukes induce a severe peritonitis with ascites. According to own observations the ascites fluid contains severe amounts of eosinophil granulocytes. Subacute fasciolosis was shown to be an important factor for reduced reproductive performance in sheep by means of high rates of non-pregnancy, reduced twinning rates and protracted lambing periods [24]. The hepatic lesions cause high levels of toxic metabolites and inflammatory mediators that impede embryonic implantation and maintenance of the early pregnancy.

Chronic fasciolosis develops more slowly after ingestion of lower numbers of metacercariae over a longer range of time, for example, if sheep graze only on a pasture with a little contamination. Mature flukes parasitize and reproduce in the bile ducts causing cholangitis and bile duct obstruction resulting in the symptoms of chronic *Fasciola* spp. infections. Clinically, sheep infected chronically demonstrate oedema (especially in the region of the lower jaw, “bottle jaw”), weakness, diarrhoea, shedding of wool and pallor due to anaemia [25]. The chronic stage of this disease is indicated by hepatomegaly and in some cases posthepatic icterus. The infected animals often show a normochrome, normocytic anaemia, accompanied by eosinophily and granulocytosis in the differential cell counts [26].

Fouda et al. [27] surveyed the influence of chronic fasciolosis on the body weight of sheep and its implication on cellular and biochemical constituents of the blood. The animals in this study exhibited progressive weight loss in adult sheep, underweight lambs, profuse diarrhoea and oedematous swelling of the intermandibular space. The blood results showed a significant decrease in the total erythrocyte cell counts, haemoglobin concentration, and packed cell volume in infected animals compared to healthy control animals. Leukocytosis and marked eosinophilia were also evident. The observed anaemia was related to the blood feeding activity of both immature and adult flukes. Consequently, the significant decrease in the levels of copper, iron, zinc, glucose, total protein and albumin in *Fasciola*-infected sheep was observed as a result from the blood loss. Moreover, fasciolosis is a predisposing factor for mastitis [28] and drops in coagulation parameters [29].

### 2.4. Diagnosis of Fasciola spp.

The classical method for the diagnosis of a *Fasciola* spp. infection is the coproscopic detection of eggs in faecal material using a sedimentation method. Another method is the use of a coproantigen ELISA, which detects *Fasciola*-antigens in the faeces up to three weeks before egg shedding begins [30]. Further possibilities for diagnosing *Fasciola* spp. infections are the detection of antibodies in serum as well as the use of molecular methods such as the polymerase chain reaction (PCR) or loop-mediated isothermal amplification (LAMP) [31]. However, each method has different advantages and disadvantages: The classical sedimentation method is inexpensive and easy to perform, but it can only diagnose patent infections and due to the intermittent egg shedding animals with low fluke burdens might be diagnosed false negative [32,33]. The coproantigen ELISA is able to detect fluke infection as early as 5–6 weeks p.i. [30], enabling a diagnosis during the prepatency. However, the costs for performing the coproantigen ELISA are much higher than those for the coproscopic method, which needs only basic laboratory equipment and tap water. The detection of specific anti-*F. hepatica*-antibodies in serum samples with an ELISA technique is a suitable method for early diagnosis of infection [34]. In experimentally infected sheep, specific antibodies were detectable within two weeks p.i. [35]. The serious disadvantage of the antibody test is the missing possibility to discriminate between active and past infections as the antibodies persist for about 12 weeks after successful treatment [36]. A study from 2016 [31] performed a comparison between conventional methods (faecal egg counting, coproantigen ELISA and serology) with molecular methods (PCR and LAMP) for diagnosing *F. hepatica* in the field. The authors conclude from their study, that the conventional methods are still more sensitive than the molecular methods for the diagnosis using faecal samples [31].

### 2.5. Treatment of Fasciola spp. and Drug Resistance

The drug of choice for the treatment of fasciolosis is triclabendazole (TCBZ) at a dose of 10 mg/kg bodyweight. TCBZ is a halogenated benzimidazole that is effective against all fluke stages; however, its exact mode of action is still unclear. Noteworthy, the widespread use of TCBZ for many years led to the development of resistant *Fasciola* populations (for review see: [11]). The first report of resistance was published in 1995 in Australia [37]. Since then, TCBZ-resistance has been reported in several countries all over the world during the last decades. Other drugs for treating *Fasciola*-infected livestock are albendazole, closantel, nitroxynil and oxyclozanide. However, all of them are only effective against the mature flukes and not against the juvenile stages.

### 2.6. Make Use of Genomic Resources to Unravel Resistance Mechanisms

Genome assemblies have been published for *F. hepatica* [38,39] and *F. gigantica* [40] and analysis of such data will be very helpful for future screening of potential drug targets, as e.g., shown by McVeigh et al. [41]. The two *F. hepatica* genome assemblies have an overall size of 1.3 and 1.14 GB, respectively [38,39]. With N50 values of 161 and 205 kbp, the quality of the assembly appears reasonably good and 90% of all core eukaryotic genes were identified in both assemblies. In both genome assemblies, a high proportion of repetitive sequences was identified, which explains the large genome size in comparison to other digenean flukes [39]. For comparison, the genome assembly of *F. gigantica* had an overall size of 1.04 GB and also has a high content of repetitive sequences. With an N50 value of 129 kbp, this genome assembly also appears to be more fragmented than the *F. hepatica* assemblies [40]. The study by Cwiklinski et al. [39] revealed that haplotype diversity is high within *F. hepatica* by resequencing individual flukes from five different UK isolates. This suggests that fluke populations can react with considerable plasticity to changes in the environment as caused by climate change and drug treatment. From a practical perspective, such genome data also help us to understand the mechanisms of resistance and in the long term helps to develop markers for rapid and cost-effective diagnosis of resistance using molecular approaches [42]. Future studies combining approaches such as clonal amplification of *F. hepatica* lines with known resistance status with whole genome resequencing using genome-wide association analyses might provide clues to how selection of resistance against flukicides such as TCBZ can be decelerated. Furthermore, this could help to develop tools for the early detection of multi-drug resistant populations and consequently optimize treatment advice. Lastly, this genome data will be valuable to guide research towards the development of effective vaccine candidates.

### 2.7. Strategies towards a Fasciola Vaccince

Apart from management changes such as fencing out water trenches and other snail habitats, vaccination would be an attractive way to control infections of *Fasciola* spp.; thereby reducing the need for treatments as well as production loss caused by the parasite. The development of vaccines has been a researched for the last decades [43,44]. However, the identification of potential vaccine antigens is a challenge due to the intricacy of the parasite’s molecules to modulate the response of the host’s immune system towards a non-protective reaction [45,46]. During an active infection, the host protective Th1 response is suppressed by a dominant, non-protective Th2 immune response [13]. This phenomenon is also observed in other chronic helminth infections, promoting the survival and the longevity of the parasites in their definitive hosts [47].

The current state of research regarding vaccinations against *Fasciola* spp. has recently been reviewed by McManus [48]. The most promising vaccine candidates so far have been various cathepsin L proteases, leucine aminopectidase, haemoglobin and peroxiredoxin. Although efficacies as high as 75–90% were observed in some trials particularly using the leucine aminopeptidase antigen [49,50,51], there was a high variability between trials. However, that might be due to the use of different adjuvants and a lower efficacy of recombinant antigens compared to that of native antigens purified from somatic fluke proteins or excretory/secretory material [48]. Current data also suggest that a combination of several antigens does not lead to a higher level of protection compared to the efficacy of a single antigen in a vaccine [48]. Despite the high variability of genomes between different *Fasciola* isolates, the variability of antigens used in vaccines was low suggesting that this does not contribute to variability in vaccination efficacy [52].

The choice of the adjuvant is highly relevant as emphasized in another very recent publication [44]. In this study, a vaccination trial using two vaccine candidates in Merino sheep was performed. The two candidates consisted of four *F. hepatica* recombinant molecules formulated in two different adjuvants (Montanide ISA 61 VG in group 1 and Alhydrogel^®^ in group 2). Although the antigen mixture was exactly the same in both vaccines, the sheep in group 1 showed a significantly lower fluke burden and a significant decrease of hepatic lesions compared to the untreated infected control group. In contrast, this was not observed in the group 2 which was immunised with the same antigens in a formulation with the aluminium-based adjuvant (Alhydrogel^®^). This points out, that the selection of the adjuvant does play a major role in terms of inducing protection against *F. hepatica* [44].

Thus, although there are promising activities regarding vaccine development against *Fasciola* spp., practical availability of such a vaccine presumably cannot be expected in a short or even medium future.

In summary, *Fasciola* spp. cause general emaciation and lower productivity in sheep due to hepatic and biliary damage and impaired metabolic pathways following from compromised liver function. The degree of severity and the clinical course is mostly dependent on the number of ingested metacercariae, which initially determines the fluke burden. Reduced appetite, decreased feed conversion efficacy, nutritional deficiency and hypoproteinaemia are the main reasons for wasting and weakness in chronically *Fasciola*-infected sheep.

## 3. *Dicrocoelium* spp. (Lancet Flukes)

Similar to the *Fasciola* spp., the lancet fluke parasitises the liver of mammalian definitive hosts, especially in grazing wild and domestic ruminants. Infections of rabbits, pigs, dogs and horses do also occur sporadically [53] and the disease has zoonotic potential as well.

### 3.1. Dicrocoelium Species and Their Global Distribution

Dicrocoeliosis, caused by *Dicrocoelium dendriticum*, the lancet fluke, is also a globally distributed parasitic disease. It was first described in 1803 in Europe and due to the movement of infected animals [54] an endemic situation is now seen in numerous countries of Europe, Asia, America and North Africa. The distribution of the other *Dicrocoelium* species, *Dicrocoelium hospes* is limited to some regions in Africa [55] and *Dicrocoelium chinesis* occurs in many regions of Asia in ruminants but has also been found in Sika deer in Europe [56]. However, other details about these trematode species are substantially similar to *D. dendriticum*.

*Dicrocoelium dendriticum* (Figure 4) is 6–12 mm in length and 1.5–2.5 mm in width presenting in a lanceolate form with a smaller oral sucker and a larger ventral sucker, which are located in close proximity. The body of the fluke is semi-transparent, so that the internal organs are visible through the external surface. In contrast to *Fasciola* spp., *D. dendriticum* does not have external spikes on its tegument [10] (p. 85). The dark-brown, thick-shelled eggs are small, measuring 35–45 µm in length and 22–30 µm in width and the operculum is often inconspicuous [10] (p. 387).

### 3.2. Life Cycle of Dicrocoelium Dendriticum

The parasite has a complex triheteroxenic life cycle with terrestrial molluscs and ants as first and second intermediate hosts. That is why *D. dendriticum* is relatively independent of moist habitats compared to *Fasciola* spp. Over 100 land snail species have been found to serve as natural and/or experimental intermediate hosts of *D. dendriticum* [57].

The adult trematodes live in the gall bladder and the bile ducts of the mammalian definitive hosts and shed their embryonated eggs via the bile to the intestine and from there mix with the faeces into the environment [56]. Snails ingest the eggs when feeding on mammalian faeces and inside the digestive tract of the mollusc, the miracidium hatches subsequently out of the eggshell (most likely due to physical-chemical stimuli [58,59]). After invading the intestinal wall of the snail, it settles in the hepatopancreas [57,60]. Two generations of sporocysts develop from the miracidia before cercariae are generated, a process that takes approximately 3–4 months [60]. This episode of the fluke’s life cycle is dependent on a temperature higher than 4 °C and the outside temperature correlates positively with the rate of development inside the snail [61]. The cercariae migrate to the respiratory tract of the snail, where they are coated with slime before being eliminated through respiratory movements [57]. Various species of ants of the family Formicidae ingest these cercariae-containing slime balls [56]. Inside the ant, the cercariae lose their tails and at least one of them reaches the suboesophageal ganglion. The remaining cercariae develop into infective metacercariae in the abdomen of the ant. The first cercaria (called “brainworm”) in the ganglion impacts the behaviour of the infected ant, manipulating the ant to cling on vegetation during the nights, which raises the chance significantly for the ant including the infectious metacercariae in the abdomen to be ingested by the grazing definitive host [62]. The number of metacercariae per ant varies. In the study from Manga-González et al. in 2001, the authors observed numbers of 2 to 240 metacercariae per ant; however, they mentioned further studies with even higher counts [57]. Inside the gut of the mammalian host, metacercariae emerge out of its cyst walls. Subsequent to the excystation, the juvenile flukes follow the route through the common bile duct to reach the liver [57]. After reaching maturity, the adult flukes start a new reproduction cycle by shedding new eggs [57]. Experimentally infected lambs showed a prepatent period of 49–79 days post infection [62].

A schematic life cycle is depicted on the website of the Center for Disease Control (USA), available online under the following link: www.cdc.gov/dpdx/dicrocoeliasis/index.html (accessed on 5 February 2021).

### 3.3. Clinical Features of Dicrocoeliosis

Due to the complexity of the life cycle of the fluke, artificial experimental infections for scientific research are difficult to implement. Therefore, published data about the pathogenic effects of *D. dendriticum* on animals are sparse.

Theodoridis et al. [63] investigated the pathophysiological implications that *Dicrocoelium*-infections had on twenty adult sheep with fluke burdens ranging from less than 100 to more than 4000 flukes by monitoring body weights, packed cell volumes, serum albumin, and serum total protein. The authors concluded that there was no significant loss of red blood cells or plasma albumin at the different levels of infection [63].

Manga-Gonzalez et al. [64] quantified the hepatic marker enzymes (ALT, AST, GGT, ALP and LDH) and other biochemical values in relation to infection doses of 1000 and 3000 *Dicrocoelium* metacercariae using twelve lambs per infection group. They also compared the body weight of the infected lambs with the weight of an uninfected control group. The authors documented a slight increase in the serum levels of albumin and bilirubin in the infected animals as well a rise in the mean activities of the liver enzymes, especially regarding ALT and AST during the early stage of infection until 60 days p.i. However, no clear correlations between the fluke burden and the alterations in biochemical parameters or in the hepatic marker enzymes were observed. During the post-mortem examination, hepatic induration, whitish dilated intrahepatic bile ducts and enlarged gall bladders were diagnosed. Concerning the body weight, *D. dendriticum* infections impacted the animal weight negatively particularly until 60 days p.i. (the estimated period until young flukes have reached maturity). In this study, the decrease in weight gain in comparison to the control group was the only clinical symptom the authors detected. The lowest weight increase was recorded 60 days p.i. (−15% in the group infected with 3000 metacercariae and −12% in the group infected with 1000 metacercariae) [64].

Other authors state that a lancet fluke burden of less than 1000 worms does not have clinical significance or economic repercussions [65,66,67]. Wolff et al. [68] found that infected sheep with 3000 metacercariae resulted in a fluke burden of up to 1946, nevertheless the infected animals did not show any clinical signs. In contrast, lambs infected with 3000 metacercariae had diarrhoea and a reduced growth rate [69]. Furthermore, Sargison et al. identified dicrocoeliosis as a probable predisposing cause of weight loss and hepatogenous photosensitisation in Scottish sheep [70]. Naturally infected animals were found to occasionally show anaemia, oedema, emaciation, and in advanced cases cirrhosis and scarring on the liver surface [53], as well as proliferation and alterations in the septal bile ducts of the lobular hepatic edges [69,71]. The lack of considerable clinical symptoms observed in patho-physiological studies was considered to be the consequence of the behaviour of the immature flukes, which migrate directly through the bile ducts without invading the intestinal wall or liver tissue of the definitive host [63]. In addition, the authors did not observe any significant loss in blood and plasma proteins under natural infections with a fluke burden of up to 4000 adult flukes.

### 3.4. Diagnosis of Dicrocoelium Infections

Unlike the eggs of the common liver fluke, the smaller eggs of *Dicrocoelium* spp. in faecal samples cannot reliably be detected using a sedimentation method [53]. In a study from 1999, a modified McMaster method using a HgI_2_ /KI solution (specific gravity 1.44) for flotation showed significantly better results than the sedimentation technique or a flotation using ZnSO_4_ solutions (specific gravity 1.3 and 1.45) or K₂CO₃ (specific gravity 1.45) [72]. Similar to *Fasciola* spp., serum samples can be examined for antibodies against *Dircocoelium* spp. with an ELISA as an alternative to coprological methods [73]. The presence of antibodies in serum can first be detected four weeks p.i. [74]. However, due to the lack of clinical symptoms at low fluke burdens, dicrocoeliosis often stay undiagnosed during the animal’s lifespan and the diagnosis is made post-mortem at the abattoir [53].

### 3.5. Treatment and Prevention of Dicrocoeliosis

Regarding the treatment of dicrocoeliosis, benzimidazoles and pro-benzimidazole derivates such as albendazole and thiophanate at higher doses than used against nematodes were shown to be effective [53,75], whereas treatment with triclabendazole did not show efficacy against *D. dendriticum* [76]. To our knowledge, there are no published reports of anthelmintic resistance for *D. dendriticum* to date. Due to the fact that many *Formica* spp. second intermediate host ant species are protected, at least in Europe [77], prevention due to management measures is particularly difficult in extensive breeding systems and land preservation projects involving extensive grazing by ruminants. An attempt to reduce the number of intermediate hosts might be by the placing of entomophagous and molluscophagous poultry such as turkeys, chickens and ducks to the grazing areas [53]. However, that method will work on small pastures only. As combating the intermediate hosts is not a reliable prophylactic strategy, an effective drug treatment of livestock is the most useful control method of dicrocoeliosis [78].

To the knowledge of the authors, there are no published data on approaches to develop vaccines against *D. dendriticum*. Due its low pathogenicity, this parasite is obviously not of high priority for such efforts.

### 3.6. Genomic Resources and Their Potential Use for Control of Dicrocoelium spp. Infections in Livestock

Data from a genome assembly of *D. dendriticum* are available in GenBank since 2014 (BioProject accession no: PRJEB3954), but they are not associated with any publication. The overall sequence length of about 550 MB is in the same range as for many foodborne trematodes [39]. However, with an N50 value of only 403 bp, the assembly appears to be highly fragmented and probably of low value in its current stage.

In conclusion, the pathogenic effects of *D. dendrititum*-infections on sheep are difficult to quantify, as sheep in regions where *D. dendriticum* is prevalent may additionally be infected with more pathogenic flukes (e.g., *F. hepatica*) or gastrointestinal nematodes or lungworms, which cause symptoms quite similar to dicrocoeliosis [63]. Artificial infection experiments are challenging to conduct due to the complexity of the life cycle of the parasite. According to existing literature it can be summarised, that *D. dendriticum* infections do not appear to be an as remarkable risk factor for chronic wasting in sheep compared to *Fasciola* spp., even at high fluke burdens. However, the infection can cause weight loss or a decreased weight gain in infected sheep [64,69,70].

## 4. Paramphistomidae as Rumen and Liver Flukes

Rumen flukes infecting ruminants belong to the family Paramphistomidae and several species within this family infect sheep, including *Calicophoron daubneyi, Cotylophoron cotylophorum*, *Paramphistomum leydeni, Paramphistomum ichikawai, Paramphistomum microbothrium, Gigantocotyle explanatum* and *Gastrothylax crumenifer*.

### 4.1. Species and Distribution of Paramphistomidae

The infestation of sheep with rumen flukes (Paramphistomidae) is widespread worldwide. In Europe, local occurrences have been reported for decades. They have been recorded in many countries and regions, such as Bulgaria, France, Poland, Hungary, Italy, India, Russia and Sardinia, and were also recorded in Yugoslavia [79]. In the last few years, it has become evident that paramphistomosis is gaining more importance in northern and central Europe, because of the high prevalence rates reported in the United Kingdom, Ireland, France, Spain, Belgium and the Netherlands. In these European countries, almost exclusively, *Calicophoron daubneyi* was diagnosed [80,81,82,83,84,85,86,87,88,89]. Only sporadically was *Paramphistomum leydeni* identified [86]. Since 2016, diagnosis of paramphistomosis have also been found to be increased in northern Germany in sheep [90]. *Paramphistomum ichikawai* is mainly found in Africa, Asia and Australia [91,92].

Worldwide, a large number of other species belonging to the genera *Gastrothylax*, *Fischoederius*, and *Calicophoron* were also recorded [93]. Their host range are domestic ruminants such as cattle, water buffalos, sheep and goats as well as wild ruminants such as deer, buffalos, etc. All these trematodes have a similar indirect life cycle. The ontogeny proceeds as a two-host cycle, similar to that of *F. hepatica*. Small terrestrial snails (*Planorbis planorbis*, *Anisus vortex*, *Bathyomphalus contortus*, etc.) serve as intermediate hosts. Like *F. hepatica*, *C. daubneyi* uses *G. truncatula* gastropods as intermediate host. The infection of sheep is oral-alimentary by ingesting the metacercariae with the grass on wet pastures or in standing or slow-floating waters in the shore zone. Under central European conditions, the main periods of infection are in late summer and autumn. After oral uptake the metacercariae leave the cyst shell in the small intestine, juvenile flukes penetrate the mucous membrane. After several days in the intestine, they migrate within the mucosa to the abomasum and duodenum and finally into the rumen. In acute paramphistomosis, the juvenile rumen flukes can remain in the duodenum for up to four weeks. Adult paramphistomes inhabit the rumen and lay clear, operculated eggs that are passed into the faeces. In sheep, the prepatency is up to 115 days [94]. After experimental infection of goat kids with *C. dabneyi*, the prepatency was estimated to be 12–13 weeks. For *C. calicophoron* a much shorter prepatency period of seven weeks was observed [95].

One paramphistome species of Asia, *Gigantocotyle explanatum*, migrates to and matures in the bile ducts [96]. This fluke is a very common amphistome in bile ducts and gall bladders of cattle and buffaloes in numerous countries, and the parasite has also been found in sheep [96]. Although it induces hyperplasia of bile duct epithelium with marked proliferation of mucosal glands and mononuclear cell infiltration, clinical symptoms are lacking [96,97].

Adult *C. daubneyi* rumen flukes are glassy-pink to flesh-coloured and have a conical, rounded shape (Figure 5). They have a size of 6–12 mm, making them visible to the unaided eye. The larger belly sucker is at the rear end.

### 4.2. Life Cycle of Paramphistomidae

The life expectancy of the flukes is considered to be several years. When faeces are excreted into water, miracidia develop from these eggs over a period of 12 to 21 days depending on temperature. Swimming miracidia enter water snails of various genera including *Planorbis*, *Bulinus*, *Lymnaea*, *Galba*, *Gyraulus* and others. Mature sporocysts containing rediae develop in snails over a period of 11 days. Rediae are released within the snail and after an additional maturation of 10 days, contain multiple cercariae. In turn, cercariae are released from rediae and mature within the snails for approximately 13 more days [94]. Mature cercariae are released from snails into water under a stimulation of strong sunlight. Liberated cercariae attach to herbage and encyst as metacercariae, remaining viable for three months. When ruminants ingest contaminated herbage, metacercariae excyst in the small intestine where they feed aggressively and mature for a period of 6–8 weeks [94]. Young flukes migrate anteriorly within the mucosa of the gastrointestinal tract to the rumen, where they undergo additional maturation after attachment to the rumen mucous, begin laying eggs, and complete its life cycle. Migration to the rumen and its subsequent development can be delayed or prolonged up to several additional months when fluke infestations are heavy [26,94].

According to experimental infections of cattle, sheep, and goats with *Paramphistomum microbothrium*, maturation of flukes from the intestine to the rumen is completed in sheep and cattle within 34 days, but just begins in goats for that time frame. Further, the subsequent egg laying begins two weeks later in goats than in cattle or sheep. The prolonged residence of maturing paramphistomes in the small intestine may contribute to increased pathogenicity of the fluke in goats [94].

### 4.3. Epidemiology and Clinical Features of Infections with Paramphistomidae

Studies on the presence of paramphistomosis in goats at different slaughterhouses in Bangladesh revealed a high prevalence of 73%, within the three species of amphistomes: *Paramphistomum cervi*, *Cotylophoron cotylophorum* and *Gastrothylax crumenifer*. Mixed infections with two or more species of amphistomes were found in 60% of the goats [98]. Abattoir examinations in Jammu, India, revealed that 36.2% of sheep and 30.9% of goats were positive for paramphistomosis. The season of the year had a significant influence on the prevalence. A higher percentage of positive animals were found in the rainy and post-rainy season as compared to the summer and winter seasons [99].

In Europe epidemiological studies on amphistomes have been carried out in Italy, where in two regions (southern Apennines and Campania), about 16% and 14% of the sheep holdings were infected and a high positive correlation with the presence of *F. hepatica* was observed [100,101]. In Wales, flock level prevalence of *C. daubneyi* for sheep (42%) was significantly lower than for cattle (59%). Co-infection with *C. daubneyi* and *F. hepatica* was observed on 46% of the farms. The presence of streams and bog habitats, and Ollerenshaw index values (an index increasing with rainfall and decreasing with extraterrestrial radiation) were significant positive predictors for the presence of *C. daubneyi* [87]. In Ireland, the rumen fluke prevalence increased in cattle in the time between 2010 to 2013 from 36.4 to 42.5%, and in sheep from 12.4 to 22%. Within the same time period, the prevalence of the liver fluke fluctuated in cattle and sheep year by year but remained always below the prevalence of the rumen flukes. In sheep, the prevalence of liver flukes was higher than in cattle [102]. A national prevalence study of rumen fluke infections was completed from November 2014 to January 2015 in Ireland. An apparent herd prevalence of 77.3% was determined. Several risk factors were identified: flocks predominantly grazing lowland pastures were more than twice as likely to be positive compared to those grazing in mountain pastures. Sharing the paddocks with other livestock species, especially cattle included an increased risk for sheep. There was also a higher susceptibility of the Irish Suffolk sheep breed to infection with rumen flukes [86]. In the Netherlands, the average flock prevalence in the years 2009–2014 was 8%. The epidemiological studies raise intriguing questions regarding a competition of *C. daubneyi* with *F. hepatica* and effects of climate change on *C. daubneyi* establishment [87].

Concerning infestation of rumen flukes, a basic distinction is made between intestinal or abomasal and ruminal paramphistomosis. The ruminal form is subclinical. In the adult stage the trematodes are relatively harmless inhabitants of the rumen (Figure 6).

However, acute and subacute paramphistomosis are observed during migration of the rumen flukes in the duodenum and abomasum and might be connected with clinical symptoms. The clinical course depends on the age of the sheep, intensity of infestation, and type of pathogen. The symptoms of paramphistomosis appear 16–32 days post infection: initially reduced appetite, apathy, later diarrhoea, mild fever, oedema, anaemia, recumbency and emaciation. After uptake of up to 50,000 metacercariae under experimental conditions, the intestinal paramphistomosis can cause a severe catarrhal enteritis [103,104].

In sheep, it was reported that the intestinal paramphistomosis can also occur in naive adults. In a small hobby flock, several adult ewes became sick and in two of them numerous young flukes were found in the small intestine at post-mortem examinations [105]. Once the infection has been overcome, an immunity is induced that largely prevents the re-colonisation and migration of rumen flukes in the intestinal mucous membrane.

Pathological-anatomical changes are to be found in the migration pathways of the trematodes in the small intestine and abomasum, as well as in the attachment sites in the rumen. The intestinal mucosa is reddened, swollen and sections of petechial haemorrhages occur. Young flukes can be found deep between the villi. The abomasal mucosa is catarrhally inflamed in the area of the pylorus [26,92].

Button-like protrusions of the rumen mucosa and villi atrophies develop at the attachment points of the abdominal suction cup. Damage to the papillary body at the attachment points can lead to atrophy and necrosis or growths of the papillary body and cellular infiltration of the submucosa [26,91].

### 4.4. Diagnosis of Rumen Flukes

Patent rumen fluke infections can be detected by means of a sedimentation method. The eggs of Paramphistomidae are similar to those of *F. hepatica*, but they are slightly larger, coarser structured and light grey, clear and operculated. In case of a suspected intestinal paramphistomosis, the faeces are washed through a fine sieve and the residue in the sieve is investigated for young flukes under the microscope [91,106]. Commercial serological tests for the detection of paramphistomid infections are not available. In rumen dissection, the conspicuous flesh-coloured trematodes are found between the rumen villi in the area of the anterior and posterior rumen pillars and at the rumen reticulum border [26]. Previously, the most common diagnostic method for differentiation of rumen fluke species was the microscopical examination. Increasingly, genome determination, sequencing, and polymerase chain reactions have become widely used methods for species identification [86,107,108].

### 4.5. Treatment and Prevention of Rumen Fluke Infections

Elimination of the immature flukes with appropriate anthelmintic therapy is the major objective and in severe cases may be lifesaving, if treatment is started early in the course of the disease.

Morantel citrate at a dose of 6 mg of morantel base/kg bodyweight (b.w.) has been shown to be 99.5% effective against immature paramphistomes. In addition, Bithionol (25–100 mg/kg b.w.), Niclofolan (6 mg/kg b.w.), Niclosamide (50–100 mg/kg b.w.), and Resorantel (65 mg/kg b.w.) are reported to be 95% effective against immature flukes in sheep and goats. Bithional may be toxic to goats at the increased effective dose ranges. Oxyclozanide (15 mg/kg b.w.) has a slightly less consistent efficacy range of 85% to 100% against immature flukes but is also 100% effective against mature stages [94]. In goats, an increased dose of 22.5 mg/kg b.w. has not a better treatment effect against juvenile flukes [109]. There are contradictory reports on the efficacy of albendazole in sheep. Sey [110] regarded albendazole at a dose of 20 mg/kg b.w. as highly effective, whereas Rolfe and Boray [92] saw nearly no effect. In dairy cattle, albendazole and netobimin had an effect of 0 to 26% in the faecal egg count reduction test. Better results were achieved with closantel and oxyclozanide with FECR values of 97–99%, so that Arias et al. [111] recommended the administration of closantel in those countries where oxyclozanide is not available.

Prophylactic measures (pasture hygiene, amelioration) are aimed at ditches and ponds, which are located on the pasture and are suitable biotopes for amphibic snails.

Since diagnosis of rumen flukes is difficult during the phase when severe symptoms can occur, vaccination would be a valuable approach to avoid such severe cases. However, little has been done in this direction except for a single study in which a recombinant *Paramphistomum epiclitum* haemoglobin has been used to vaccinate calves, however, no protective effect was observed [112].

### 4.6. Available Omics Data on Rumen Fluke

While no genome data are available, there are two publications describing transcriptome data for *Paramphistomum cervi* [113] and *Calicophoron daubneyi* [114]. Since the latter also provides data on differential gene expression between freshly excysted juveniles and adult flukes and proteomic data on the secretome using mass spectrometry, this data set will be very valuable in the future to identify candidate antigens for systematic vaccine development projects.

## 5. Conclusions

Although liver and rumen flukes can cause considerable clinical disease, the current options for control are limited, e.g., due to drug resistance (*Fasciola*) or poor diagnostic options in the phase of the most severe clinical burden (Paramphistomidae). Since the diseases often take a chronic course, wasting is a frequently observed outcome, particular if animals face high infection pressure, additional infections, or if stress factors aggravate the problem. A better understanding of the biology of these parasites will help to improve intervention strategies and sheep health.

## Figures and Tables

**Figure 1 animals-11-00549-f001:**
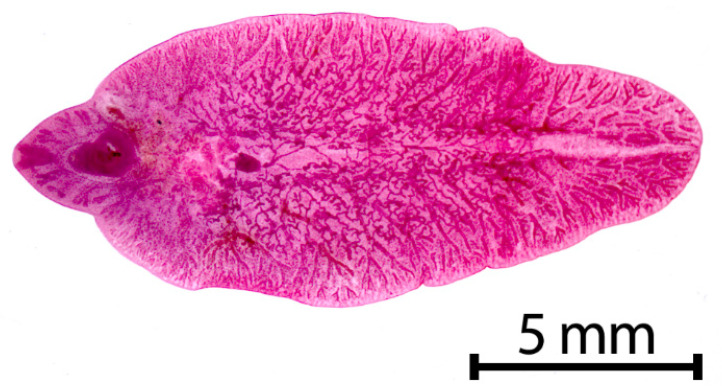
Adult stage of *Fasciola hepatica*.

**Figure 2 animals-11-00549-f002:**
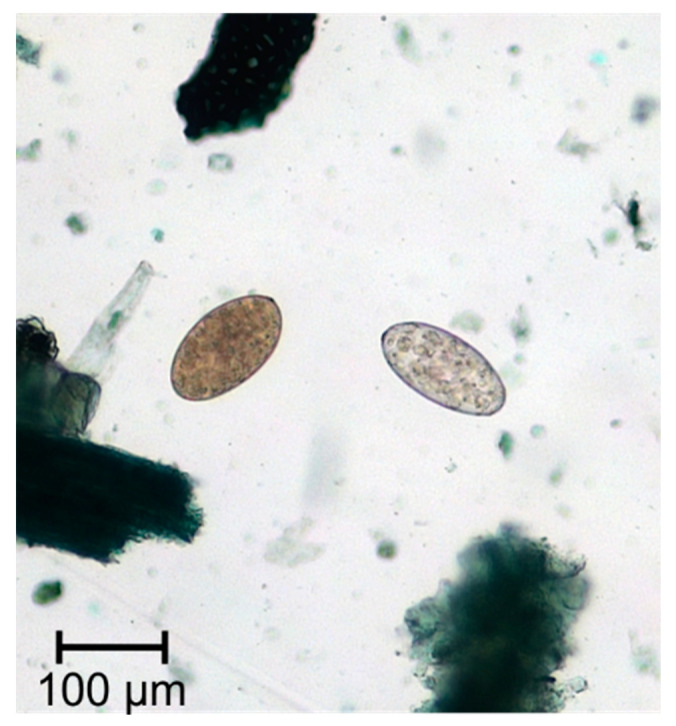
Eggs of *Fasciola* spp. and *Paramphistomum* spp.

**Figure 3 animals-11-00549-f003:**
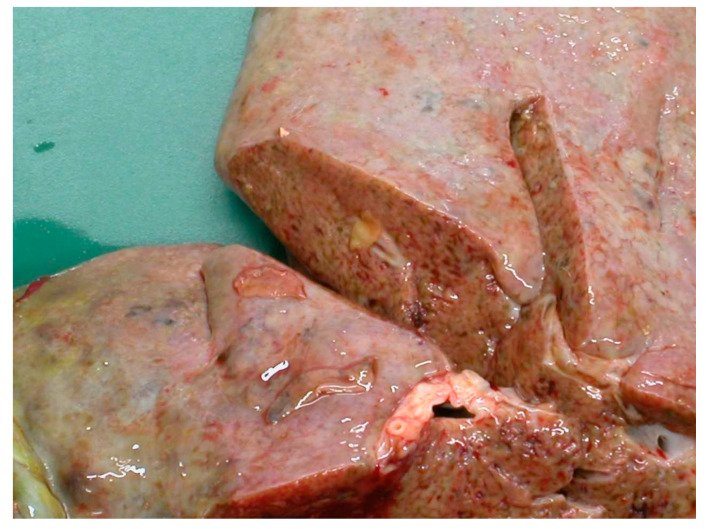
Hepatic damage caused by the migration of the juvenile flukes.

**Figure 4 animals-11-00549-f004:**
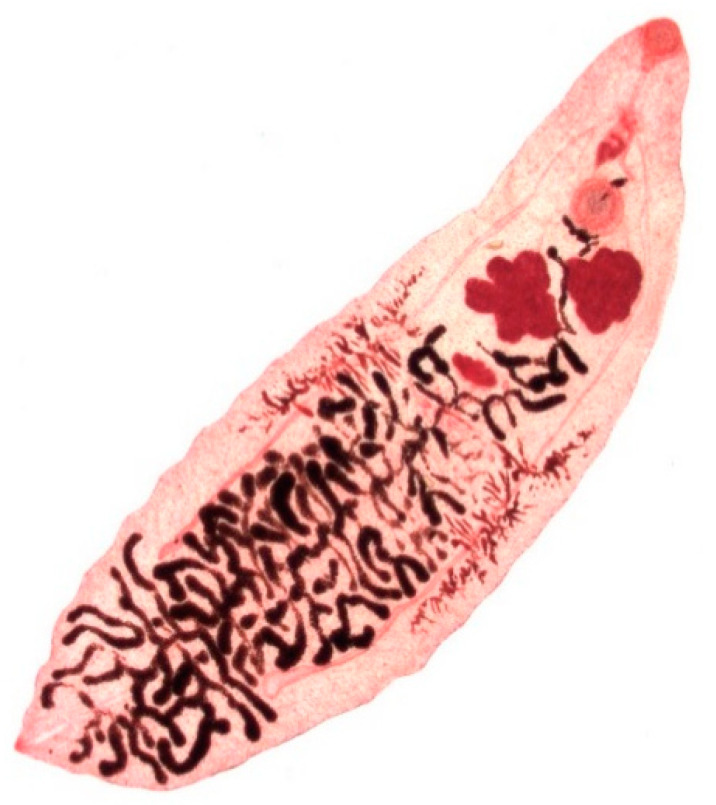
Adult stage of *Dicrocoelium dendriticum*.

**Figure 5 animals-11-00549-f005:**
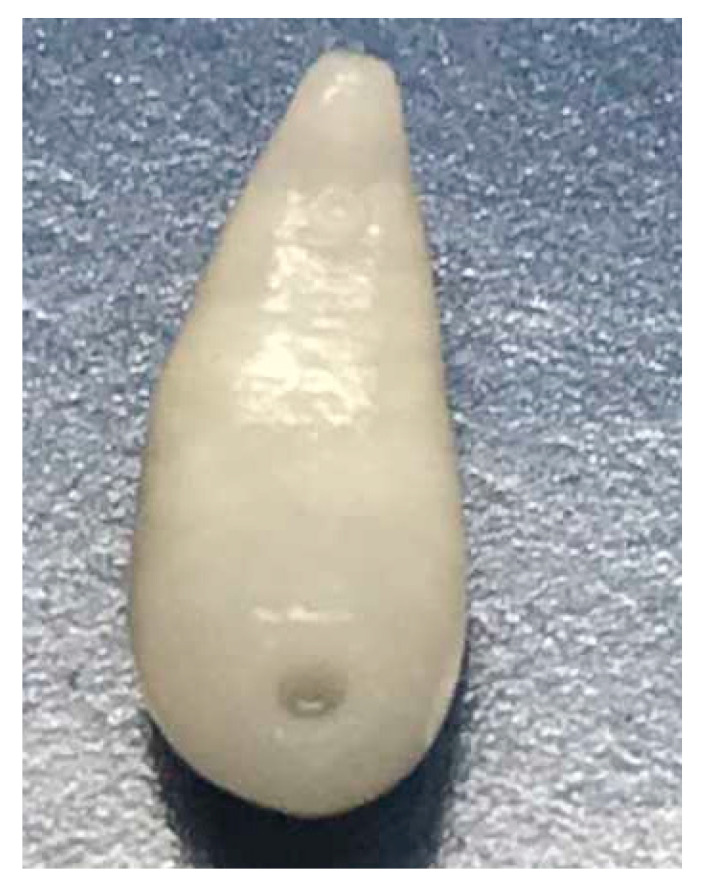
Adult *Calicophoron daubneyi* rumen fluke.

**Figure 6 animals-11-00549-f006:**
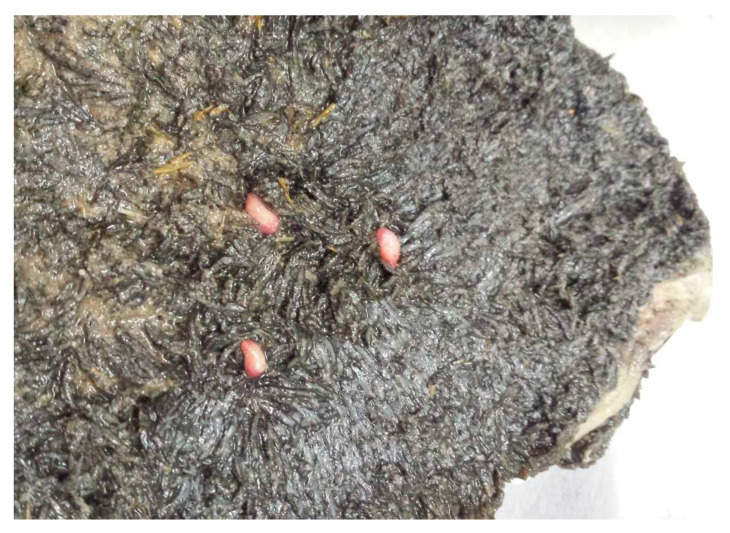
Adult flukes parasitizing in the rumen.

## Data Availability

Data sharing not applicable.

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
