# Peer review of "Chronic Wasting Due to Liver and Rumen Flukes in Sheep"

_animals, 2021, doi:10.3390/ani11020549_

Round 1
Reviewer 1 Report
The review is interesting and illustrative. However, programs for the prevention of parasitic diseases and the use of vaccines are little discussed. There is published information on the genome of the parasites and the expression of genes in the animal. I suggest including a section on these topics in the manuscript. The conclusion must be more relevant.
Reviewer 2 Report
It is well written, covers the trematodes of importance, and that it is a good bibliographic review on the subject.As it is not an experimental research work, I cannot contribute anything new to what has already been written by the authors.
The bibliography consulted is new, and the works reviewed are correct.
I regret that I cannot provide any suggestions on this work.
Reviewer 3 Report
This is the simple review paper about trematods of ruminats. The strengh of the manuscript is the fact that authors based mainly on well selected and relatively new references (literature).
I think that the schematic life cycles of described flukes would enrich this review and make it well understandably for readers
Reviewer 4 Report
Revisions for Animals-1073151
The manuscript “Chronic Wasting due to Liver and Rumen Flukes in Sheep” provides information on flukes’ species in sheep. The review is well-conducted and interesting. English language should be revised to correct some minor errors to facilitate understanding. Some changes in the structure are needed for a better organization.
The manuscript should be considered for publication in Animals, following minor revisions.
Here my commentaries:
Please, consider dividing each paragraph related to the different flukes’ species, in sub-paragraph, e.g., morphology, life cycle, epidemiology, clinical disease, diagnosis, treatment, etc., since the manuscript results confusing in some points.
Please, check and revise the use of italics for the name of parasites in all the manuscript and in figure captions.
Line 219: “However, each method has different advantages and disadvantages.” Details and references are needed.
A conclusion paragraph should be provided.
